# Peripubertal Alterations of Leptin Levels in Patients with Autism Spectrum Disorder and Elevated or Normal Body Weight

**DOI:** 10.3390/ijms24054878

**Published:** 2023-03-02

**Authors:** Katarzyna E. Skórzyńska-Dziduszko, Agata Makarewicz, Anna Błażewicz

**Affiliations:** 1Department of Human Physiology, Medical University of Lublin, 11 Radziwiłłowska Street, 20-080 Lublin, Poland; 2Department of Psychiatry, Psychotherapy and Early Intervention, Medical University of Lublin, 20-439 Lublin, Poland; 3Department of Pathobiochemistry and Interdisciplinary Applications of Ion Chromatography, Medical University of Lublin, 1 Chodźki Street, 20-093 Lublin, Poland

**Keywords:** leptin, autism spectrum disorder, obesity, puberty

## Abstract

Leptin, which plays a key role in energy homeostasis, is known as a neurotrophic factor possibly linking nutrition and neurodevelopment. Available data on the association between leptin and autism spectrum disorder (ASD) are confusing. The aim of this study was to explore whether plasma levels of leptin in pre- and post-pubertal children with ASD and/or overweightness/obesity differ from those of BMI- and age-matched healthy controls. Leptin levels were determined in 287 pre-pubertal children (mean age 8.09 years), classified as follows: ASD with overweightness/obesity (ASD+/Ob+); ASD without overweightness/obesity (ASD+/Ob−); non-ASD with overweightness/obesity (ASD−/Ob+); non-ASD without overweightness/obesity (ASD−/Ob−). The assessment was repeated in 258 of the children post-pubertally (mean age 14.26 years). There were no significant differences in leptin levels either before or after puberty between ASD+/Ob+ and ASD−/Ob+ or between ASD+/Ob− and ASD−/Ob−, although there was a strong trend toward significance for higher pre-pubertal leptin levels in ASD+/Ob− than in ASD−/Ob−. Post-pubertal leptin levels were significantly lower than pre-pubertal levels in ASD+/Ob+, ASD−/Ob+, and ASD+/Ob− and higher in ASD−/Ob−. Leptin levels, elevated pre-pubertally in the children with overweightness/obesity as well as in children with ASD and normal BMI, decrease with age, in contrast to the increasing leptin levels in healthy controls.

## 1. Introduction

Leptin is a proinflammatory cytokine, primarily synthesized by white adipose tissue. This adipokine is secreted in direct proportion to the amount of white adipose tissue, and therefore, the plasma level of leptin is an indicator of the body’s energy stores and caloric intake [1]. Leptin plays an important role in energy homeostasis via appetite-suppressing and neuroendocrine effects mediated by hypothalamic receptors [1,2]. A complex network of interacting signaling pathways of leptin and insulin appears to regulate food intake, energy balance, and body weight [1,2]. In addition to signaling in the central nervous system, leptin exerts its metabolic effects by acting directly in peripheral tissues [3]. Increased leptin levels are linked to obesity and metabolic syndrome. There is also evidence that leptin is produced by the brain, where it is postulated to act as a paracrine and/or autocrine regulator; however, the role of brain-derived leptin remains unclear [4]. Emerging evidence suggests that leptin alters brain structure, neuron excitability, and synaptic plasticity and regulates feeding circuits, suggesting that it may play a role in the development of the central nervous system [5,6,7]. Leptin signaling is thought to play a critical role in hippocampus-dependent learning through regulation of synaptic plasticity and neurotransmitter receptor trafficking [8]. Moderate concentrations of leptin have been shown to improve memory, whereas high concentrations impair memory and synaptic transmission [9]. Leptin is also a critical neurotrophic factor, and leptin signaling abnormalities during fetal development have been associated with decreased neuronal stem cell differentiation and growth [10].

Several recent studies have shown the possible role of leptin and leptin receptors in the development of neurodegenerative diseases via the modulation of neuroinflammation, dopaminergic neurodegeneration, and neuroprotection [11,12,13]. Furthermore, a postmortem study with brain tissue showed higher concentrations of leptin, together with a number of proinflammatory and modulatory cytokines, in the anterior cingulate gyrus of patients with autism spectrum disorders (ASD) than in that of controls [14]. Autism spectrum disorders are a group of phenotypically diverse neurodevelopmental syndromes that manifest early in development. They include a number of developmental brain disorders with a wide range of symptoms and levels of impairment, characterized by communication difficulties, social impairment, and repetitive or stereotyped behaviors [15].

Several cross-sectional studies have investigated the association between circulating levels of leptin and ASD [16,17,18,19,20,21,22,23,24]; however, the available data are confusing. In some studies plasma levels of leptin were shown to be higher in autistic children of normal or elevated body weight compared with children without ASD [16,17,18,20,21,22,23]. In contrast, Prosperi et al. did not identify any significant correlation between the levels of leptin and the presence or absence of a regression of skills prior to the onset of autism in 85 preschoolers with ASD [24].

Despite the originality and novelty of the cited studies, the interpretation of the data should take into account some limitations. The studies show relatively weak associations between leptin and ASD. Furthermore, except the pilot study by Dhaliwal et al., all of the studies suggesting an association of leptin with ASD were conducted in populations of normal-weight children, without controlling for the effect of BMI on leptin in the context of ASD along with co-existing overweight or obesity. Since higher rates of obesity and overweight continue to be reported among children with ASD compared with the general population [25,26], such studies should be designed to take into account elevated body weight. Thus, further BMI-controlled, drug-naïve human studies are needed to examine leptin levels in the context of ASD.

Furthermore, to the best of our knowledge the alterations of leptin levels in children with ASD have not been studied longitudinally in relation to puberty; however, there are several prospective studies focusing on the alterations of leptin over extended periods of time [27,28,29]. Since maternal overweight and obesity are well-known risk factors for autism in children [30,31], a few longitudinal studies addressing leptin levels in the perinatal period and early childhood in the context of ASD have been conducted [27,28,29]. Raghavan at al. showed that extremely rapid weight gain during infancy and elevated leptin levels during early childhood were independently associated with greater ASD risk in later years [27]. Interestingly, no associations were found between birth weight for gestational age or cord leptin and risk of ASD [27]. Similarly, Young et al. reported that there was no significant positive association between maternal second-trimester leptin levels and ASD [28]. In contrast, Iwabuchi et al. found significant associations between cord serum leptin levels and increased autistic symptoms [29].

As demonstrated above, a number of studies have explored the possible link between leptin and ASD, but the available results are confusing or contradictory. To our knowledge, none of the previous studies has focused on peripubertal evaluation of leptin levels in terms of both ASD and elevated body weight. In view of this research gap, the aim of this study was to explore whether plasma levels of leptin in pre- and post-pubertal children with ASD differ from those of healthy controls, in the wider context of an elevated BMI, as well as its peripubertal changes.

### Study and Control Population

The study was conducted on a carefully chosen population diagnosed with ASD. The diagnosis was based on the criteria for autistic disorder as defined in the Diagnostic and Statistical Manual of Mental Disorders, Fifth Edition (DSMV), Autism Diagnostic Interview Revised (ADI-R), and Autism Rating Scale (CARS) [15].

Inclusion criteria included pre-pubertal age and ASD diagnosis. Patients were recruited through local support groups (mainly parents and caregivers) or were referred by specialist clinicians and therapists. The informed consent of the children’s parents or legal guardians was obtained. The control group (neurotypical children and adolescents) was selected from among children visiting the medical center for regularly scheduled check-ups. They were not receiving any medical treatment. All members of the control and study groups were from the same geographic area (central and south-eastern Poland).

Exclusion criteria included the presence of any chronic condition, drug treatment for any acute or chronic condition, and refusal to give informed consent.

An initial personal interview was conducted by trained personnel to establish household characteristics, dietary behavior, health status, and socioeconomic characteristics. Each child’s weight status was determined using an age- and sex-specific percentile for body mass index (BMI, in kg/m^2^). Current and representative BMI percentile charts for the Polish population of children and adolescents (3–18 years of age) were used [32]. BMI was divided into two categories representing normal weight or elevated weight (both overweight and obesity).

A total of 287 pre-pubertal children (mean age: 8.09 years; standard deviation: 1.36) were recruited into the study and divided into four groups: autistic patients with overweight or obesity (ASD+/Ob+); autistic patients without overweight or obesity (ASD+/Ob−); non-autistic patients with overweight or obesity (ASD−/Ob+); and non-autistic patients without overweight or obesity (ASD−/Ob−).

The clinical assessment was repeated after normal puberty in a total of 258 children (mean age 14.26 years, standard deviation 1.37) from the four groups (Figure 1). The study combined data from the 2014/2015 and 2019/2020 cycles of measurements to provide more statistically reliable estimates. Both girls and boys were carefully examined for signs of puberty. The gold standard for establishing pubertal onset is physical examination with palpation of breast bud development and testicular enlargement for girls and boys, respectively. All the post-pubertal children enrolled in the post-pubertal phase of the research had commenced puberty at least one year before the post-pubertal clinical assessment and blood sample collection. All females got their first period, and all the males began to have ejaculation at night before the post-pubertal clinical assessment. Children who did not present pubertal onset within the required time frame, as well as those who had developed a chronic condition affecting their weight or limiting their ability to participate in the study or had started any chronic pharmacological treatment, were not enrolled in the second phase of the research.

Pre-pubertally, a small group of study participants were diagnosed as underweight (*n* = 39) according to the above-mentioned BMI percentile charts (for age and gender) for the Polish population of children and adolescents [32]. BMI values below −1 SD for girls, as well as BMI values less than −1.5 SD for boys, were considered underweight.

The pre-pubertally underweight male group (*n* = 34; 24 boys from the ASD+/Ob group; 10 boys from the ASD−/Ob− group) was characterized as follows:
mean BMI 13.42 kg/m^2^; standard deviation 0.69; median 13.53 kg/m^2^; interquartile range 0.9; minimum value 11.9 kg/m^2^; maximum value 14.69 kg/m^2^;mean age 7.39 years; standard deviation 1.14; median 7.92 kg/m^2^; interquartile range 2.0; minimum value 6 years; maximum value 10 years.

According to the Polish BMI percentile chart, being underweight in a male at the age of 7.39 years is considered to be below 13.8 kg/m^2^.

The pre-pubertally underweight female group (*n* = 5; 1 girl from the ASD+/Ob− group; 4 girls from the ASD−/Ob− group) was characterized as follows:
mean BMI 13.4 kg/m^2^; standard deviation 0.74; median 13.23 kg/m^2^; interquartile range 1.11; minimum value 12.6 kg/m^2^; maximum value 14.35 kg/m^2^;mean age 7.15 years; standard deviation 1.01; median 7.83 kg/m^2^; interquartile range 1.75; minimum value 6 years; maximum value 8 years.

According to the Polish BMI percentile charts, being underweight in a female at the age of 7.15 years is considered to be below 14 kg/m^2^.

All pre-pubertally underweight children presented normal physical development pre- and post-pubertally, and their weight after puberty had normalized spontaneously. The mean difference between the normal BMI cut-offs (stratified for age and gender) and BMI calculated in the underweight group was 0.43 kg/m^2^. Therefore, we decided to include this group in the study. The single outliers in the data were BMI 11.96 kg/m^2^ for a 6-year-old boy (the ASD+/Ob− group) and BMI 11.9 kg/m^2^ for a 6-year-old boy (the ASD−/Ob− group).

According to the classification of ASD severity (DSMV), the overwhelming majority of the recruited autistic patients presented a mild level of interference in functioning and support required (the level 1). Very few recruited children with ASD presented a moderate level (the level 2).

## 2. Results

### 2.1. Characterization of Study Groups and Wilcoxon Signed-Rank Test Results of the Comparison of Related Samples before and after Puberty

A detailed characterization of the children divided into four groups is presented in Table 1. As the variables were not normally distributed and nonparametric median tests were used to compare samples, the mean and standard deviation values are presented only to provide a full characterization of the study groups. Table 1 also presents the Wilcoxon signed-rank test results of the comparison of related samples before and after puberty.

Interestingly, the Wilcoxon signed-rank test results of the comparison of leptin levels before and after puberty revealed that they were significantly lower after puberty in three groups, i.e., ASD+/Ob+, ASD+/Ob−, and ASD−/Ob+, as follows:
ASD+/Ob+ group (*p* < 0.001): pre-pubertal median (Me) = 24.7 ng/mL, interquartile range (IQR) = 13.1 versus post-pubertal Me = 21.85 ng/mL, IQR = 18.5ASD+/Ob− group (*p* < 0.001): pre-pubertal Me = 5.6 ng/mL, IQR 5.76 versus post-pubertal Me = 4.6 ng/mL, IQR = 4.8ASD−/Ob+ group (*p* = 0.001): pre-pubertal Me = 26.7 ng/mL, IQR = 10.1 versus post-pubertal Me = 22.4 ng/mL, IQR = 14.44

In contrast, after puberty, leptin levels were significantly higher (*p* < 0.001) only in the ASD−/Ob− group (healthy control group): pre-pubertal Me = 2.86 ng/mL, IQR 1.25 versus post-pubertal Me = 4.73 ng/mL, IQR 2.62.

### 2.2. Kruskal–Wallis ANOVA Test Results of the Comparison of Four Groups (ASD+/Ob+, ASD+/Ob−, ASD−/Ob+, and ASD−/Ob−)

All the groups (ASD+/Ob+, ASD+/Ob−, ASD−/Ob+, and ASD−/Ob−) were age-matched before and after puberty.

There were no significant differences in pre- or post-pubertal BMI values between the two groups of patients with overweightness/obesity (ASD+/Ob+ and ASD−/Ob+) or between the two groups with normal body weight (ASD+/Ob− and ASD−/Ob−); however, a comparison of the two groups of overweight/obese pre-pubertal children revealed a distinct trend (*p* = 0.071) toward significance: ASD+/Ob+ group (median (Me) = 25.71 ng/mL, interquartile range (IQR) = 4.7) versus ASD−/Ob+ (Me = 22.69 ng/mL, IQR = 2.19). The trend was not observed after puberty. The extended analysis of the subgroups (ASD+/Ob+, ASD+/Ob−, ASD−/Ob+, and ASD−/Ob−), additionally stratified for gender, confirmed the lack of statistical significance for BMI between the respective groups. Furthermore, the above-mentioned trend toward significance, observed in the comparison of the two groups of overweight/obese pre-pubertal children, was not present in subgroups of females and males.

Notably, there were no significant differences, either pre- or post-puberty, in leptin levels between the two groups of patients with overweightness/obesity (ASD+/Ob+ and ASD−/Ob+) or between the two groups with normal body weight (ASD+/Ob− and ASD−/Ob−) (Figure 2 and Figure 3), although the comparison of non-ASD and ASD lean pre-pubertal children revealed a distinct trend toward significance (*p* = 0.15, higher leptin levels in the ASD+/Ob− group than in the ASD−/Ob− group) (Figure 2).

### 2.3. Kruskal–Wallis ANOVA Test Results of the Comparison of Eight Groups Stratified by the Direction of BMI Changes Presented by Children after Puberty in Comparison to Their Pre-Pubertal BMI

The comparative analyses of leptin levels were also conducted in the subgroups stratified for the direction of BMI changes presented by children after puberty in comparison to their pre-pubertal BMI values (Figure 4). There were no significant differences in leptin levels between the groups of patients with and without ASD with pre-pubertal overweight/obesity and sustained overweight/obesity after puberty (ASD+/Ob+/Ob+ versus ASD−/Ob+/Ob+) (Figure 4 and Table 2). The leptin levels in these two groups were significantly higher than in the other tested groups, which included children with or without ASD who had normalized BMI after puberty; children with or without ASD with normal weight pre-pubertally but overweight/obesity after puberty; and children with or without ASD with normal weight both before and after puberty (Figure 4). Very surprisingly, there were no significant differences in leptin levels between children with and without ASD with normal weight both before and after puberty or between children with and without ASD whose BMI had changed after puberty (i.e., who had normalized BMI or had presented increased BMI values after puberty). Multiple comparisons of mean ranks of leptin levels for eight groups stratified by the direction of BMI changes presented by children after puberty in comparison to their pre-pubertal BMI are presented in Table 2. This lack of significant differences in leptin levels between children with normal weight, both before and after puberty, and children whose BMI value had changed after puberty was observed despite the significant differences in BMI values between these groups. The BMI changes are presented in Figure 4 and Table 3.

### 2.4. Regression Analyses of Leptin Serum Levels Associations

In the multiple linear regression analyses (Table 4), leptin serum levels showed no significant relationships with the presence of ASD. Leptin serum levels were positively associated with BMI values.

### 2.5. Results of the Comparative Analyses of Leptin Serum Levels in the Subgroups Stratified for Gender

The extended analysis of the subgroups (ASD+/Ob+, ASD+/Ob−, ASD−/Ob+, and ASD−/Ob−), stratified for gender, revealed no statistical significance in median leptin levels between females and males with overweight/obesity either before or after puberty (the ASD+/Ob+ and ASD−/Ob+ groups) (Table 5). Interestingly, in all the groups with normal body weight (ASD+/Ob− and ASD−/Ob−, before and after puberty) leptin levels were significantly higher in females than in males (Table 5), despite the lack of significant differences in the BMI values of the respective subgroups.

## 3. Discussion

### 3.1. Leptin Levels in Autistic Patients with Normal Weight Compared to Healthy Controls

In the present study we examined the possible association between leptin and ASD before and after puberty. Our findings show that there were no significant differences in leptin levels either before or after puberty between lean patients with ASD (ASD+/Ob−) and age-matched healthy controls (ASD−/Ob−), although the comparative analysis revealed a strong trend toward significance (*p* = 0.15) for higher leptin levels in the pre-pubertal ASD+/Ob− group (median = 5.6 ng/mL) than in the ASD−/Ob− group (median = 2.86 ng/mL). This lack of significance did not result from the lower median BMI value in the ASD+/Ob− group (14.23 kg/m^2^) relative to the ASD−/Ob− group (15.99 kg/m^2^), as the difference between these two medians was not statistically significant (*p* = 0.3). Interestingly, in post-pubertal children, the trend for higher leptin levels was no longer observed (ASD+/Ob− group, median = 4.6 ng/mL versus ASD−/Ob− group, median = 4.76 ng/mL), and the median BMI values were almost equal in both groups.

Several cross-sectional studies have previously shown a positive association between circulating levels of leptin and ASD [16,17,18,19,20,21,22,23]. The trend of higher leptin levels in children with ASD and normal body weight is in agreement with the results of those studies; however, an interpretation of the available data should take into account some limitations. Due to the relatively small sample sizes in these studies, as well as in our research, a generalization of the findings is difficult.

There are also several serious methodological differences between these studies and the present study. For example, Ashwood et al. found that leptin levels were significantly higher in age-matched children with autism (70 children) compared with typically developing non-ASD controls (50 age-matched children): median = 2.11 ng/mL versus 0.96 ng/mL, *p* < 0.006 [16]. There were no statistical differences in BMI between autism cases (median = 16.79 kg/m^2^) and controls (median = 16.68 kg/m^2^) [16]. Since Ashwood’s study group included children whose age varied widely between 2 and 15 years, and median leptin levels in this population were markedly lower than those obtained in our research, the results reported by Ashwood et al. cannot be directly compared to our data [16].

Rodrigues et al. reported that plasma levels of leptin were significantly increased (*p* < 0.01) in 30 autistic children (median = 1.691 ng/mL) compared with 19 children without ASD (median = 0.862 ng/mL) [18]. Despite its novelty, the cited research seems to present some limitations: 1/no data were given on either age or BMI values, although the authors stated that they had controlled for the effect of BMI; 2/the reported level of significance (*p* < 0.01) was close to the threshold of significance; 3/full statistics were not presented in the study [18].

Similarly, another case-control study reported significantly increased leptin levels (*p* ≤ 0.01) in a small group of 31 lean pre-pubertal children with autism (ages ranged from 3 to 8 years with mean ± SD = 5.59 ± 2.26 years) compared to 28 healthy age- and sex-matched controls [20]. Blardi et al. observed elevated plasma leptin over one year in 35 young normal-weight post-pubertal autistic patients (mean age at the basal time 14.1 ± 5.4 years), compared to 35 healthy sex- and age-matched subjects [17]. In a study by Çelikkol Sadıç et al., which included a total of 44 (38 boys, 6 girls) children with ASD and 44 (35 boys, 9 girls) healthy controls aged 18–60 months, plasma leptin levels were reported to be significantly higher (*p*  <  0.001) in the ASD group (2.91  ±  0.94 ng/mL) than in the control group (2.27  ±  0.66 ng/mL); however, the BMI percentile was also found to be higher in children with ASD than in the control group (*p*  =  0.027) [21]. Interestingly, no relationship was found between the severity of ASD symptoms, severity of eating problems, and plasma levels of leptin [21]. In a study by Maekawa et al., leptin level trajectories showed a borderline-significant difference between the ASD (*n* = 123) and control (*n* = 92) children at 4–12 years of age only in terms of the Spearman’s correlation coefficients (*p* = 0.0499) [22]. In contrast, a recent report by Prosperi et al. indicated no significant correlation between leptin levels and the presence or absence of a regression of skills prior to the onset of autism in 85 preschoolers with ASD [24].

In conclusion, a comparative analysis of the results is nearly impractical due to the limitations of the research, including relatively small sample groups in all studies; highly varying median age between studies—from very young children to much older post-pubertal teenagers; failure to control for the effect of BMI on leptin levels; weak associations that manifested as borderline *p*-values in several of the studies; different ranges of leptin levels in most of the studies; very small—but reported as significant—differences between median values of leptin between autistic and neurotypical children in several studies; the absence of data on the use of psychotropic medication in most of the studies; and possible differences between ethnic populations. A well-powered study will be helpful for conclusive confirmation of a positive relationship between leptin levels and ASD in pre-pubertal children with normal weight. This association was revealed in the present study as a strong trend toward significance for higher leptin levels in the ASD+/Ob− group than in the ASD−/Ob− group. The lack of significance in our report was not due to the relatively lower median BMI value in the ASD+/Ob− group (14.23 kg/m^2^) compared to the ASD−/Ob− group (15.99 kg/m^2^), as the difference was not statistically significant *p* = 0.3. In post-pubertal children, in whom the trend for higher leptin levels was no longer observed, the median BMI values were almost equal.

Interestingly, the present study documented for the first time that with increasing age and after the onset of puberty, the difference in leptin levels between ASD and non-ASD children with normal weight disappears. Furthermore, we observed for the first time that leptin levels were significantly lower after puberty in the ASD+/Ob− group (*p* < 0.001): pre-pubertal median = 5.6 ng/mL versus post-pubertal median = 4.6 ng/mL. In contrast, in the healthy control group (ASD−/Ob−) leptin levels were significantly higher (*p* < 0.001) after puberty (pre-pubertal median = 2.86 ng/mL versus post-pubertal median = 4.73 ng/mL).

A post-pubertal increase in leptin levels in healthy subjects is a physiological phenomenon observed during normal puberty. Leptin levels rise gradually with age, prior to puberty, suggesting that a threshold effect may trigger puberty [33,34]. Emerging reports suggest that leptin plays a role in adrenarche, as leptin plasma levels increase with higher levels of body fat, and leptin can modulate both the hypothalamus–pituitary–adrenal axis (HPA) and the hypothalamus–pituitary–gonadal axis (HPG) [35,36]. Importantly, distribution of body fat, particularly gluteofemoral fat, is profoundly associated with start of menarche, more than the amount of total body fat [36,37,38,39]. In boys, leptin plausibly has a minor impact in adrenarche, although androgen levels are suggested to be positively associated with plasma leptin levels in several studies [36,40].

Leptin levels exhibit sexual dimorphism, which is independent of body mass index. Females have higher leptin levels than males due to an increase in leptin expression in subcutaneous adipose tissue, stimulation of leptin synthesis by estrogen, and inhibition of leptin synthesis by testosterone [3,41]. As in previous research [41,42], in our study, the groups with normal body weight (ASD+/Ob− and ASD−/Ob−, before and after puberty) presented significantly higher leptin levels in females than in males despite the lack of significant differences in BMI values between these groups.

To conclude, on the basis of the present and previous findings on children with normal body weight, it seems plausible to assume that leptin levels may play a much more important role at birth and early childhood than in later years of adolescence in ASD etiopathology. An association between elevated cord serum leptin levels and later ASD symptoms was shown in 762 Japanese children aged 8–9 years [29]. However, two other studies conducted in much smaller sample groups did not confirm this association [27,28].

Finally, we conclude that leptin levels, elevated pre-pubertally in children with ASD and normal BMI, tend to normalize with age, and after puberty, they do not differ from leptin levels in healthy controls.

### 3.2. Leptin Levels in Patients with ASD and Overweightness/Obesity Compared to Neurotypical Children with Overweightness/Obesity

According to current estimates of the World Health Organization, more than one billion people worldwide are obese—650 million adults, 340 million adolescents, and 39 million children. This number is still increasing [43,44]. Higher rates of obesity and overweightness continue to be reported among children with ASD compared with the general population [25,26]. Children with ASD have been observed to have an over 41% greater risk of developing obesity [26]. Emerging evidence suggests that the elevated risk for obesity among children with ASD is attributed to additional, specific risk factors, in addition to typical obesity risk factors [23,25,45]. These specific risk factors include psychotropic medication use, medical comorbidities, and the presence of specific genetic factors [23,25,45]. It was also reported that extremely rapid weight gain during infancy may be associated with a greater risk of ASD development in childhood [27].

In our study, there were no significant differences in BMI values between the two groups of patients who are overweight/obese (ASD+/Ob+ and ASD−/Ob+) either before or after puberty; however, the comparison of these two groups before puberty revealed a distinct trend (*p* = 0.071) toward significance for higher BMI values in the ASD+/Ob+ group (median = 25.71 ng/mL) than in the ASD−/Ob+ group (median = 22.69 ng/mL). This trend was not observed after puberty. Since the exclusion criteria in our study included the presence of any chronic condition or chronic drug treatment, the role of these additional, specific risk factors in the development of obesity is probably minimized. This may be reflected by the lack of significance between the two groups of patients who are overweigh/obese (ASD+/Ob+ and ASD−/Ob+) in our research.

Several studies have documented weight gain in children with ASD treated with olanzapine, risperidone, and aripiprazole [46,47]. Chronic risperidone exposure in children with autism causes weight gain in excess of developmentally expected norms, which follows a curvilinear trajectory and decelerates over time [48]. In a recent cross-sectional analysis by Srisawasdi et al., a total of 168 children and adolescents with ASD treated with a risperidone-based regimen for ≥12 months presented leptin dysregulation in a dose- and duration-dependent manner [49]. Another study in 97 children with a mean of 22.9 ± 2.8 weeks of risperidone exposure showed a weight gain of 5.4 ± 3.4 kg weight gain over 24 weeks (*p* < 0.0001), an increase in waist circumference from 60.7 ± 10.4 cm to 66.8 ± 11.3 cm (*p* < 0.0001), and significant increases in leptin levels (*p* < 0.0001) [50]. In contrast, Esen-Danaci et al. found decreased levels of leptin in patients treated with risperidone [51].

As several studies have documented a positive correlation of leptin levels with BMI in non-autistic children [52], as well as in children with ASD (*p* = 0.002) [24], it remains unclear whether the increase in leptin levels is a secondary mechanism to the weight gain induced by psychotropic medication use or whether psychotropic medication also acts directly on the synthesis and/or turnover of leptin to modulate its concentration. Despite the originality and novelty of the previously cited studies on the role of leptin in ASD [16,17,18,19,20,21,22,23,24], due to the lack of information on psychotropic medication use in several of these studies (except three studies mentioned below), the potential impact of the possible use of such medicines on BMI values and leptin levels is unclear. Although Rodrigues et al. mentioned psychotropic medication use in ASD children, these data were not included in the results of their statistical analysis [18]. In two other studies, patients with a history of psychiatric medication usage were excluded [21,24].

To the best of our knowledge, the present study is the first longitudinal drug-naïve research designed to investigate a possible association between serum leptin levels and ASD in pre- and post-pubertal children while excluding the impact of psychotropic medication on data collected from patients with elevated BMI values. Previously, we used the same model of research to examine selenium content before and after puberty in euthyroid children diagnosed with ASD compared to age-matched neurotypical controls with respect to overweight or obesity as a co-existing pathology [53]. In the present study, we documented for the first time that there were no significant differences in leptin levels between overweight/obese patients with ASD (ASD+/Ob+) and neurotypical children (ASD−/Ob+), either before or after puberty. Furthermore, we observed for the first time that leptin levels were significantly lower after puberty in the ASD+/Ob+ group (*p* < 0.001), with pre-pubertal median = 24.7 ng/mL versus post-pubertal median = 21.85 ng/mL, as well as in the ASD−/Ob+ group (*p* = 0.001): pre-pubertal median = 26.7 ng/mL versus post-pubertal median = 22.4 ng/mL. In the extended statistical analysis of children who are overweight/obese, stratified for gender, we revealed a lack of statistical significance for median leptin levels between females and males both before and after puberty (while females with normal weight presented higher leptin levels than males).

In the multivariate analyses in our study, serum leptin levels showed no significant relationships with the presence of ASD, but they were positively associated with BMI values in concordance with a previous report documenting a positive correlation between leptin levels and BMI in children with ASD (*p* = 0.002) [24].

To the best of our knowledge, the alterations of leptin levels in children with ASD and overweight/obesity have recently been studied only by Dhaliwal et al. [23]. These authors suggested that overweight/obese participants diagnosed with ASD (*n* = 6) had higher leptin concentrations (*p* < 0.02) than normal weight children with ASD (*n* = 15) [23]; however, no comparison was conducted between overweight/obese children with ASD and overweight/obese neurotypical children in that report.

A consideration of our findings in the light of existing evidence suggests that a plausible explanation for the decreasing leptin levels in both groups (ASD+/Ob+ and ASD−/Ob+) after puberty could be that males present increasing testosterone levels, which are well known to have a negative effect on leptin concentrations [3,41,54]. In males who are overweight/obese, the decreasing leptin levels that occur with increasing age and maturation may result from the gradually rising levels of testosterone. We assume that during maturation, through the complex interaction between leptin, increasing sexual hormones, and other hormonal factors, leptin levels tend to decrease despite increasing age. Interestingly, the differences between pre-pubertal and normally growing post-pubertal BMI values are much smaller in overweight/obese children (in the ASD+/Ob+ group, there was no difference at all) than in healthy controls with normal body weight.

In conclusion, pre-pubertal leptin levels in overweight/obese children are highly elevated as a result of the highly increased content of adipose tissue in relation to age. Since the amount of adipose tissue is the strongest and most important mechanism regulating the leptin level [41,42,55], other mechanisms that may influence leptin levels, including age, gender, and ASD, are suppressed. During maturation, via the triggering of specific hormonal changes, high leptin levels in overweight/obese children may begin to decrease, in contrast to the increasing leptin levels in healthy control children with normal weight. The dynamics and intensity of this alteration should be assessed in further research. 

The limitation of our research is that we did not assess plasma leptin levels in correlation with plasma concentration of reproductive hormones. Our study was not originally designed to track leptin levels in relation to the increasing levels of reproductive hormones or to the Tanner staging phases. Since reproductive hormone levels may highly vary even between healthy individuals classified in the same phase of the Tanner staging [56,57,58], and leptin levels are also modulated by co-existing ASD and/or elevated body weight, the recruited population must have been numerous enough (and much more numerous than our study group) to achieve sufficient statistical power for the reliable assessment of leptin level changes in relation to reproductive hormone levels in different stages of the Tanner scale for both genders, in the wider context of ASD and/or elevated body weight. Given the interesting link between leptin and ASD in the context of hormonal changes associated with puberty, further studies are needed to support our conclusions and to better clarify the mechanisms involved.

### 3.3. Leptin Levels in the Subgroups of Children Stratified for the Direction of BMI Changes after Puberty in Comparison to Their Pre-Pubertal BMI Values

We found no significant differences in leptin levels between the groups of patients with and without ASD with pre-pubertal overweight/obesity and sustained overweight/obesity after puberty. The leptin levels in these two groups were significantly higher than in the other tested groups (which showed no significant differences in leptin levels between them), which included children with and without ASD who had normalized BMI after puberty; children with and without ASD with normal pre-pubertal weight who presented as overweight/obese after puberty; and children with and without ASD with normal weight both before and after puberty.

A lack of association between leptin and BMI alterations has previously been reported by Antunes et al., who presented no relationship between BMI variation at 6 months and leptin [41]; nevertheless, there are also conflicting older data that describe a decrease in leptin levels after a five-week weight reduction program [59].

## 4. Materials and Methods

### 4.1. Blood Samples

Blood samples from the cubital vein were collected in the morning after overnight fasting. The study material collected during the two assessment periods consisted of 555 serum samples. The blood samples were centrifuged at 1000× *g* for 10 min, and plasma fractions were immediately stored at −80 °C until used for measurements. The leptin plasma levels were measured by an enzyme-linked immunoassay using a commercial kit (R&D systems, Minneapolis, MN, USA) according to the manufacturer’s instructions.

### 4.2. Statistical Analyses

Descriptive statistics were produced for the overall sample and also stratified by both autism and BMI status. As the Kolmogorov–Smirnov and Lilliefors tests indicated that the variables were not normally distributed, the nonparametric Kruskal–Wallis ANOVA test was used for continuous variables. The Wilcoxon signed-rank test was used to compare related samples before and after puberty. A multiple linear regression analysis was used to evaluate the association between the serum leptin levels and ASD status, BMI, and gender. All analyses were two-tailed with a significance level of 0.05 and a power of 80%. Statistical analyses were performed using TIBCO Software Inc. (2017) Statistica, version 13.0.0.0 (TIBCO, Tulsa, OK, USA), licensed to the Medical University of Lublin (used by Katarzyna Skórzyńska-Dziduszko).

## 5. Conclusions

We found no significant differences in leptin levels, either before or after puberty, between the two groups of patients who were overweight/obesity (ASD+/Ob+ and ASD−/Ob+) or between the two groups with normal body weight (ASD+/Ob− and ASD−/Ob−), although the comparison of non-ASD and ASD pre-pubertal children with normal body weight revealed a strong trend toward significance for higher leptin levels in the ASD+/Ob− group. This trend is consistent with previous findings indicating this association in children with normal body weight [16,17,18,19,20,21,22,23]; the various levels of statistical significance in the present and cited studies may result from relatively small sample sizes, as well as methodological differences between all the studies.

For the first time we found that leptin levels, elevated pre-pubertally in children with ASD and normal BMI, tend to normalize with increasing age, and after puberty they do not differ from leptin levels in healthy controls. Furthermore, we observed for the first time that leptin levels were significantly lower after puberty in the ASD+/Ob+ group as well as in the ASD−/Ob+ group. This may result from the gradually increasing levels of testosterone, which are known to have a negative effect on leptin concentrations [3,41,54]. As in previous research [41,42], in our study the groups with normal body weight (ASD+/Ob− and ASD−/Ob−, before and after puberty) presented significantly higher leptin levels in females than in males, despite the lack of significant differences in BMI values in the respective groups. This association was not observed in overweight/obese children. As the amount of the adipose tissue is the strongest and most important mechanism regulating the leptin level [41,42], other mechanisms that may influence leptin levels, including age, gender, and ASD, are suppressed in overweight/obese children.

To the best of our knowledge, the present study is the first longitudinal drug-naïve research designed to investigate a possible association between leptin levels in serum and ASD in pre- and post-pubertal children while excluding the impact of psychotropic medication on data collected from patients with normal or elevated BMI values. Our finding of decreasing leptin concentrations after puberty in all groups except the healthy control is particularly important, as it may provide a basis for further investigation to clarify the role of leptin in ASD. Further studies are needed to investigate the much more important role of leptin in childhood than in adolescence in ASD etiopathology; however, the research should take into account the informative bias resulting from elevated body weight.

## Figures and Tables

**Figure 1 ijms-24-04878-f001:**
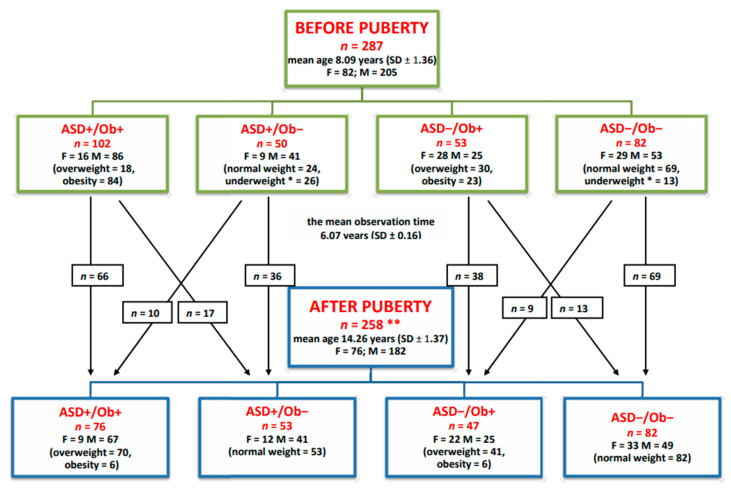
Allocation of pre- and post-pubertal children to groups. F—female, M—male. * Children diagnosed as mild underweight (near-normal weight) who presented normal physical development and normalized weight after puberty. ** A total of 29 children did not participate in the study after puberty.

**Figure 2 ijms-24-04878-f002:**
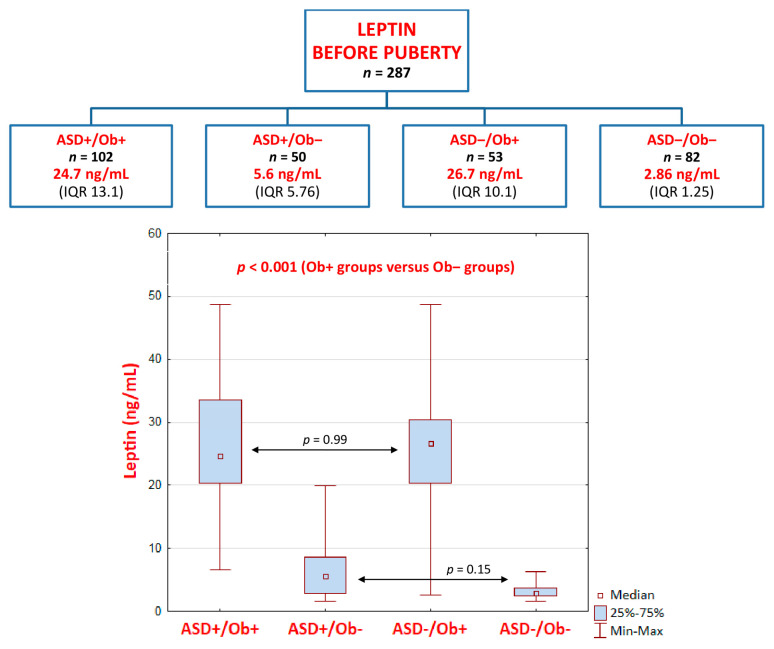
Leptin levels BEFORE puberty—the Kruskal–Wallis ANOVA test results. Leptin levels are presented as median values with interquartile ranges (IQR).

**Figure 3 ijms-24-04878-f003:**
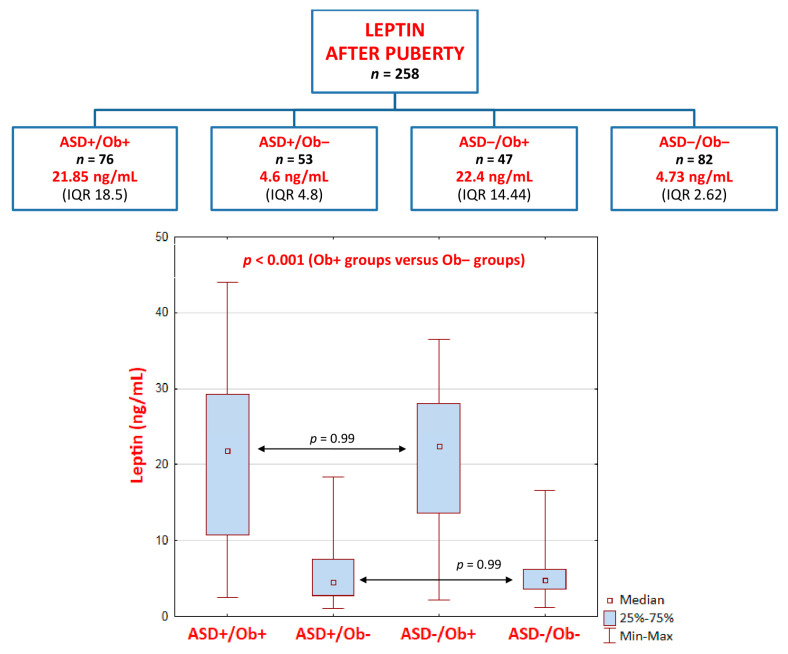
Leptin levels AFTER puberty—the Kruskal–Wallis ANOVA test results. Leptin levels are presented as median values with interquartile ranges (IQR).

**Figure 4 ijms-24-04878-f004:**
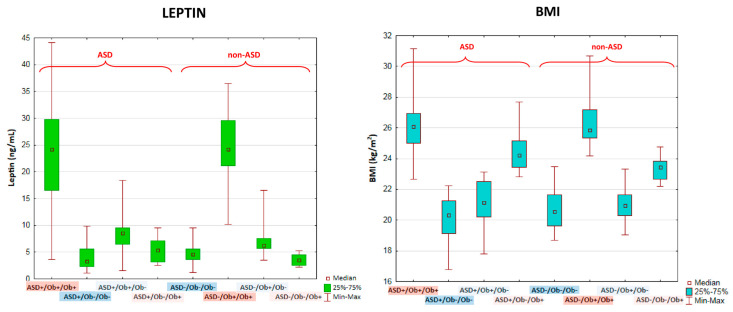
Kruskal–Wallis ANOVA test results of the comparison of leptin levels (left panel) and BMI values (right panel) in eight groups stratified by the direction of BMI changes presented by children after puberty in comparison to their pre-pubertal BMI. The data are presented as median values with interquartile ranges (IQR). The codes of groups were as follows: ASD+/Ob+/Ob+ (ASD children with pre-pubertal overweight/obesity and sustained overweight/obesity after puberty, *n* = 66); ASD+/Ob+/Ob− (ASD children with pre-pubertal overweight/obesity and normalized BMI after puberty, *n* = 17); ASD+/Ob−/Ob+ (ASD children with normal pre-pubertal weight who presented overweight/obesity after puberty, *n* = 10); ASD+/Ob−/Ob− (ASD children with normal weight both before and after puberty, *n* = 36); ASD−/Ob+/Ob+ (non-ASD children with pre-pubertal overweight/obesity and sustained overweight/obesity after puberty, *n* = 38); ASD−/Ob+/Ob− (non-ASD children with pre-pubertal overweight/obesity who had normalized BMI after puberty, *n* = 13); ASD−/Ob−/Ob+ (non-ASD children with normal pre-pubertal weight who presented overweight/obesity after puberty, *n* = 9); ASD−/Ob−/Ob− (non-ASD children with normal weight both before and after puberty, *n* = 69).

**Table 1 ijms-24-04878-t001:** Descriptive statistics of pre- and post-pubertal children and Wilcoxon signed-rank test results of the comparison of related samples before and after puberty. BMI—body mass index; min—minimum value; max—maximum value; IQR—interquartile range; SD—standard deviation.

	Mean	Median	Min	Max	IQR	SD	Wilcoxon ← Test →	Mean	Median	Min	Max	IQR	SD
	Before Puberty							After Puberty					
	ASD+/Ob+												
Age (years)	7.9	8.0	6.0	10.58	2.67	1.44	*p* < 0.001	14.07	14.25	12.08	16.67	2.71	1.46
Weight (kg)	43.14	40.85	23.9	70.9	17.7	12.34	*p* < 0.001	67.87	67.0	48.0	98.0	14.0	11.44
Height (cm)	128.13	128.35	100.0	155.0	21.0	13.04	*p* < 0.001	161.17	160.0	140.0	000	15.5	11.26
BMI (kg/m^2^)	25.71	25.27	17.99	35.16	4.7	3.35	*p* = 0.09	25.96	25.7	22.66	31.14	2.32	1.83
Leptin (ng/mL)	26.97	24.7	6.55	48.7	13.1	9.08	*p* < 0.001	21.018	21.85	2.56	44.1	18.5	10.8
	ASD+/Ob−												
Age (years)	7.75	8.21	6.0	10.33	2.25	1.28	*p* < 0.001	14.09	14.25	12.08	17.0	2.42	1.42
Weight (kg)	23.94	23.0	15.0	36.0	10.0	5.87	*p* < 0.001	52.35	52.0	38.0	78.0	13.0	9.28
Height (cm)	126.89	127.5	107.5	149.0	17.0	11.09	*p* < 0.001	159.26	159.0	141.0	185.0	13.0	10.13
BMI (kg/m^2^)	14.63	14.23	11.96	18.11	2.98	1.61	*p* = 0.01	20.47	20.66	16.76	23.12	2.15	1.59
Leptin (ng/mL)	6.19	5.6	1.56	19.9	5.76	3.87	*p* < 0.001	5.35	4.6	1.11	18.36	4.8	3.34
	ASD−/Ob+												
Age (years)	8.17	8.0	6.08	10.75	2.83	1.43	*p* < 0.001	14.08	13.5	12.17	16.5	2.17	1.26
Weight (kg)	36.51	36.1	26.7	49.0	9.0	5.33	*p* < 0.001	66.06	67.0	51.0	89.0	12.0	8.52
Height (cm)	126.66	128.0	110.0	150.0	18.0	10.49	*p* < 0.001	159.59	158.0	144.0	180.0	13.0	9.18
BMI (kg/m^2^)	22.75	22.69	19.11	27.77	2.18	1.94	*p* < 0.001	25.89	25.46	22.19	30.67	2.18	2.07
Leptin (ng/mL)	25.56	26.7	2.6	48.7	10.1	11.31	*p* = 0.001	20.25	22.4	2.16	36.5	14.44	9.92
	ASD−/Ob−												
Age (years)	8.49	8.29	6.0	10.83	1.42	1.17	*p* < 0.001	14.65	14.67	12.08	16.8	1.5	1.24
Weight (kg)	28.75	28.9	17.5	45.5	7.5	6.11	*p* < 0.001	55.38	55.0	41.0	75.0	9.0	8.12
Height (cm)	133.15	134.0	109.0	150.0	10.0	8.72	*p* < 0.001	163.05	163.0	147.0	187.0	16.0	9.98
BMI (kg/m^2^)	16.01	15.99	11.9	21.34	2.7	1.82	*p* < 0.001	20.73	20.66	18.7	23.46	1.83	1.22
Leptin (ng/mL)	3.17	2.86	1.56	6.26	1.25	1.06	*p* < 0.001	5.31	4.73	1.25	16.56	2.62	2.37

**Table 2 ijms-24-04878-t002:** Multiple comparisons of mean ranks of leptin levels for eight groups stratified by the direction of BMI changes presented by children after puberty in comparison to their pre-pubertal BMI. Kruskal–Wallis test: *H* (7, *n*= 258) = 182.45, *p* < 0.001. The codes of groups were as follows: ASD+/Ob+/Ob+ (ASD children with pre-pubertal overweight/obesity and sustained overweight/obesity after puberty, *n* = 66); ASD+/Ob+/Ob− (ASD children with pre-pubertal overweight/obesity and normalized BMI after puberty, *n* = 17); ASD+/Ob−/Ob+ (ASD children with normal pre-pubertal weight who presented overweight/obesity after puberty, *n* = 10); ASD+/Ob−/Ob− (ASD children with normal weight both before and after puberty, *n* = 36); ASD−/Ob+/Ob+ (non-ASD children with pre-pubertal overweight/obesity and sustained overweight/obesity after puberty, *n* = 38); ASD−/Ob+/Ob− (non-ASD children with pre-pubertal overweight/obesity who had normalized BMI after puberty, *n* = 13); ASD−/Ob−/Ob+ (non-ASD children with normal pre-pubertal weight who presented overweight/obesity after puberty, *n* = 9); ASD−/Ob−/Ob− (non-ASD children with normal weight both before and after puberty, *n* = 69).

ASD+/Ob+/Ob+	ASD+ /Ob+/Ob+	ASD+ /Ob−/Ob−	ASD+ /Ob+/Ob−	ASD+ /Ob−/Ob+	ASD− /Ob−/Ob−	ASD− /Ob+/Ob+	ASD /Ob+/Ob−	ASD− /Ob−/Ob+
		*p* < 0.001	*p* = 0.004	*p* < 0.001	*p* < 0.001	*p* = 1.0	*p* = 0.007	*p* < 0.001
ASD+/Ob−/Ob−	*p* < 0.001		*p* = 0.07	*p* = 1.0	*p* = 1.0	*p* < 0.001	*p* = 0.33	*p* = 1.0
ASD+/Ob+/Ob−	*p* = 0.004	*p* = 0.07		*p* = 1.0	*p* = 0.69	*p* = 0.003	*p* = 1.0	*p* = 0.41
ASD+/Ob−/Ob+	*p* < 0.001	*p* = 1.0	*p* = 1.0		*p* = 1.0	*p* < 0.001	*p* = 1.0	*p* = 1.0
ASD−/Ob−/Ob−	*p* < 0.001	*p* = 1.0	*p* = 0.69	*p* = 1.0		*p* < 0.001	*p* = 1.0	*p* = 1.0
ASD−/Ob+/Ob+	*p* = 1.0	*p* < 0.001	*p* = 0.003	*p* < 0.001	*p* < 0.001		*p* = 0.005	*p* < 0.001
ASD−/Ob+/Ob−	*p* = 0.007	*p* = 0.33	*p* = 1.0	*p* = 1.0	*p* = 1.0	*p* = 0.005		*p* = 0.9
ASD−/Ob−/Ob+	*p* < 0.001	*p* = 1.0	*p* = 0.41	*p* = 1.0	*p* = 1.0	*p* < 0.001	*p* = 0.9	

**Table 3 ijms-24-04878-t003:** Multiple comparisons of mean ranks of BMI values for eight groups stratified by the direction of BMI changes presented by children after puberty in comparison to their pre-pubertal BMI. Kruskal–Wallis test: *H* (7, *n*= 258) = 199.63, *p* < 0.001. The codes of groups are as in Table 2.

	ASD+ /Ob+/Ob+	ASD+ /Ob−/Ob−	ASD+ /Ob+/Ob−	ASD+ /Ob−/Ob+	ASD− /Ob−/Ob−	ASD− /Ob+/Ob+	ASD− /Ob+/Ob−	ASD− /Ob−/Ob+
ASD+/Ob+/Ob+		*p* < 0.001	*p* < 0.001	*p* = 1.0	*p* < 0.001	*p* = 1.0	*p* < 0.001	*p* = 0.61
ASD+/Ob−/Ob−	*p* < 0.001		*p* = 1.0	*p* = 0.001	*p* = 1.0	*p* < 0.001	*p* = 1.0	*p* = 0.06
ASD+/Ob+/Ob−	*p* < 0.001	*p* = 1.0		*p* = 0.3	*p* = 1.0	*p* < 0.001	*p* = 1.0	*p* = 1.0
ASD+/Ob−/Ob+	*p* = 1.0	*p* = 0.001	*p* = 0.3		*p* = 0.004	*p* = 1.0	*p* = 0.2	*p* = 1.0
ASD−/Ob−/Ob−	*p* < 0.001	*p* = 1.0	*p* = 1.0	*p* = 0.004		*p* < 0.001	*p* = 1.0	*p* = 0.15
ASD−/Ob+/Ob+	*p* = 1.0	*p* < 0.001	*p* < 0.001	*p* = 1.0	*p* < 0.001		*p* < 0.001	*p* = 0.49
ASD−/Ob+/Ob−	*p* < 0.001	*p* = 1.0	*p* = 1.0	*p* = 0.2	*p* = 1.0	*p* < 0.001		*p* = 1.0
ASD−/Ob−/Ob+	*p* = 0.61	*p* = 0.06	*p* = 1.0	*p* = 1.0	*p* = 0.15	*p* = 0.49	*p* = 1.0	

**Table 4 ijms-24-04878-t004:** Multivariate linear regression analyses of leptin serum levels.

Multivariate Linear Regression					
*n* = 258	BETA	SE ^a^ of BETA	B	SE ^a^ of B	*p*-Value
Leptin serum levels (ng/mL) Model adjusted for age and gender					
ASD ^b^ (ASD 1, no ASD 2)	−0.067	0.03	−1.43	0.7	0.042
BMI ^c^ (kg/m^2^)	0.71	0.065	2.45	0.22	<0.001

^a^ SE—standard error; ^b^ ASD—autism spectrum disorder; ^c^ BMI—body mass index.

**Table 5 ijms-24-04878-t005:** Kruskal–Wallis test results of the comparisons of mean ranks of leptin levels in males and females.

	Males			*p*-Value ← *p* →	Females		
	*n*	Median Leptin Level (ng/mL)	Interquartile Range		*n*	Median Leptin Level (ng/mL)	Interquartile Range
ASD+/Ob+							
Before puberty	86	23.4	13.8	not significant	16	27.82	4.25
After puberty	67	23.2	15.3	not significant	9	25.3	10.68
ASD−/Ob+							
Before puberty	25	25.3	8.0	not significant	28	29.2	7.65
After puberty	25	21.4	23.56	not significant	22	22.45	5.65
ASD+/Ob−							
Before puberty	41	4.5	3.96	*p* < 0.001	9	11.5	3.94
After puberty	41	3.5	3.07	*p* < 0.001	12	8.2	2.49
ASD−/Ob−							
Before puberty	53	2.56	0.92	*p* < 0.001	49	4.25	1.71
After puberty	29	4.12	1.06	*p* < 0.01	33	5.8	1.67

## Data Availability

The authors confirm that the data supporting the findings of this study are available within the article.

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
