# Peer review of "Peripubertal Alterations of Leptin Levels in Patients with Autism Spectrum Disorder and Elevated or Normal Body Weight"

_ijms, 2023, doi:10.3390/ijms24054878_

Round 1
Reviewer 1 Report
The article is devoted to the study of the actual topic of the relationship between leptin and autism in a cohort study.
The article needs to be improved.
My comments:
1. The authors indicated that they did not assess the severity of autism. I consider this a disadvantage. It was necessary to assess the severity of autism and then, perhaps, the results would have been different. I believe that the authors have information about the severity of autism and they will be able to recalculate the results. This is important because the authors, even in the Abstract of the article, indicate very contradictory and confusing information about the relationship between leptin and autism.
2. The authors did not assess the level of sex hormones in adolescents after the passage of puberty. Perhaps not all adolescents experienced this period in the same way. It is necessary to evaluate the levels of estrogen in girls and testosterone in boys, and then standardize the results for these parameters.
3. The introduction is very long. It needs to be improved and made more compact.
4. More than 70% of literature sources older than 5 years. It is necessary to update literary references, if possible, remove sources more than 20 years old.
Author Response
Revision Letter
Manuscript ID: ijms-2067151
Peripubertal alterations of leptin levels in patients with autism spectrum disorder and elevated or normal body weight
We would like to thank all reviewers for the intensive work on our manuscript, the critical analysis and the good recommendations. It was your valuable and insightful comments that led to improvements in the current version. We tried our best to address all criticisms, questions and comments point by point and explain how we have dealt with them (please, see the attached file with our response). In order to better follow the changes made to the manuscript, we used the "Track Changes" feature of Microsoft Word.
The authors welcome further constructive comments or questions, if any.
We would like to thank you once again for your efforts.
Yours sincerely,
Katarzyna Skórzyńska-Dziduszko MD, PhD, on behalf of the other authors

Reviewer 2 Report
The manuscript by Skórzyńska-Dziduszko et al. aimed to study the levels of leptin during the peripubertal alterations and body weight changes in patients with autism spectrum disorder (ASD). This is such a novel and important study exploring the longitudinal relationship of leptin in children’s puberty while controlling body weight. They found the leptin was increased in prepubertal children with ASD and normal weight, and decreased in obese groups after puberty. These findings will greatly improve our understanding in using leptin as a biomarker of body energy store and caloric intake to benefit the ASD patients. Such a comprehensive study around the leptin levels during peripubertal changes and body weight changes in normal and ASD children will bring important insights and guidance for improving the treatments in this area.
Here lists a few of my concerns that could improve the overall quality of the manuscript if addressed appropriately.
Q1: Line 92: Remove “with p-values close to the borderline of significance in a majority of cases”. Statistical significance indicates if a result is significant or not according to its criteria. There is no point in indicating if it’s close to the borderline or not. We should trust any result with proper statistical analysis.
Q2: Table 1: On the left, all BMI (kg/m2) should be fully showing and aligning well with the other categories.
Q3: Figure 2 and Figure 3: The titles need to be presented differently to show the focus of the figures. It’s not easy and clear for the readers to find out that Figure 2 is BEFORE puberty and Figure 3 is AFTER puberty. There’s no point to yellow highlighting the p-value in the graphs.
Q4: Figure 4: Missing titles for the two graphs, one is for Leptin, and the other is for BMI.
Author Response

(The authors gave the same response as above.)

Reviewer 3 Report
This is a well conducted, well analyzed study with largely negative, unremarkable results. As such, it helps clarify certain contradictory findings in the literature and deserves publication. There are several minor English corrections needed: line 101-best of our; line 416-impractical not impracticable; line 608- correct bogy. Review References for consistency in title capitalization.
Author Response

(The authors gave the same response as above.)

Round 2
Reviewer 1 Report
The authors of extremely dangerous articles are corrected. I have no more comments.